# Long-term trends in the intensity and relative toxicity of herbicide use

Andrew R. Kniss[1]

Herbicide use is among the most criticized aspects of modern farming, especially as it relates to genetically engineered (GE) crops. Many previous analyses have used flawed metrics to evaluate herbicide intensity and toxicity trends. Here, I show that herbicide use intensity increased over the last 25 years in maize, cotton, rice and wheat. Although GE crops have been previously implicated in increasing herbicide use, herbicide increases were more rapid in non-GE crops. Even as herbicide use increased, chronic toxicity associated with herbicide use decreased in two out of six crops, while acute toxicity decreased in four out of six crops. In the final year for which data were available (2014 or 2015), glyphosate accounted for 26% of maize, 43% of soybean and 45% of cotton herbicide applications. However, due to relatively low chronic toxicity, glyphosate contributed only 0.1, 0.3 and 3.5% of the chronic toxicity hazard in those crops, respectively.

[1] Department of Plant Sciences, University of Wyoming, Laramie, Wyoming 82071, USA. Correspondence and requests for materials should be addressed to A.R.K. (email: akniss@uwyo.edu).

Herbicides are a powerful weed control tool relied upon by farmers around the world. In the United States, herbicides accounted for 65% of all pesticide expenditures, costing farmers roughly $5.1 billion in 2007 (ref. 1). Even the most ardent critics of pesticides recognize their importance, but a perceived over-reliance on herbicides for weed control has sparked debate on how to best incorporate herbicides into sustainable weed management systems[2–4]. As Zimdahl[5] wrote: 'Whether one likes [herbicides] or deplores them, they cannot be ignored. To ignore them is to be unaware of the opportunities and problems of modern weed management.'

Regular evaluation of herbicide use is a valuable exercise, especially with respect to how herbicide use has changed over time, and whether those changes have positively or negatively affected human and environmental health. To address the reasonable desire for information about whether pesticide use changes are positive or negative, several researchers have attempted to quantify overall pesticide impacts using simplified measures; however, the two most commonly used metrics (weight of pesticides applied and the environmental impact quotient) can result in misleading or incorrect conclusions. The total amount of herbicide applied per unit area, for example, has been reported in several recent publications[2,6]. However, when a variety of different herbicides are applied, each with different use rates and toxicity profiles, simply reporting the weight of pesticide applied is dubious at best, and misleading at worst. A recent National Academy of Sciences report strongly recommended against using such non-risk assessment based approaches: 'Researchers should be discouraged from publishing data that simply compares total kilograms of herbicide used per hectare per year because such data can mislead readers'[7].

The environmental impact quotient (EIQ) developed by Kovach et al.[8] is another commonly used metric, and purports to provide an overall assessment of risk from various pesticides[6,9–11]. The EIQ combines risk factors for several aspects of toxicity and environmental health, and thus is perceived as a simple tool for comparing herbicides. However, the EIQ is a poor indicator of risk, especially for herbicides[12,13]. Due to the way toxicity data are scaled in EIQ calculations, it can readily lead to nonsensical conclusions[14]. For example, the EIQ suggests the water used to dilute and apply pesticides will typically have a greater negative environmental impact than even the most toxic herbicides[12]. Additionally, a single proxy for exposure (application timing) can explain over 25% of the variability in herbicide EIQ, even though application timing has no consistent effect on actual risk[13]. Clearly, better measures of toxicity and risk are necessary if we are to judge herbicide use in a meaningful way.

Many aspects of herbicide use are of potential interest to concerned individuals (including human health, weed resistance to herbicides, environmental risks, etc.). However, to limit the complexity of data presentation while still presenting useful, relevant information, I have chosen two aspects of herbicide use to analyze and present in-depth. The objective of this analysis was to quantify changes in herbicide use patterns in GE and non-GE crops over the last 25 years in the United States as they relate to (1) the number of herbicides being applied, and (2) the relative toxicity of the herbicides that are being used. Herbicide use intensity, as measured by the number of herbicide applications being made to each field, has increased in all crops analyzed, regardless of whether they were GE or non-GE. Even so, chronic and acute toxicity hazard associated with herbicides has remained constant or even declined in many cases.

## Results

**Increasing herbicide use.** There were a total of 159 unique herbicides in the USDA-NASS data set, but many of these entries were different formulations of the same herbicide active ingredient. For example, there were eight different salts of 2,4-D and seven different salts of glyphosate. Combining different formulations, there were 118 unique herbicide active ingredients in the full data set; 75 were used in maize, 54 in cotton, 57 in soybean, 34 in rice, 44 in spring wheat and 56 in winter wheat (Supplementary Data Set 1).

Herbicide area-treatments (roughly defined as the number of times one herbicide was applied to one field) were calculated to quantify trends in herbicide use intensity. Crop area estimates were obtained from USDA-NASS, and are provided in Supplementary Data Sets 2 through 7. It is possible (common, in fact) for the total number of area-treatments to exceed 1 (or 100% of total crop area). For example, a value of 2 area-treatments could be obtained in several ways; either by applying two different herbicides at full rates in a tank-mixture to the same field $(1 + 1 = 2)$, or by applying the same herbicide to the same field twice $(1 \times 2 = 2)$, or even by applying four different herbicides at half of their average application rates to the same field $(0.5 + 0.5 + 0.5 + 0.5 = 2)$.

A steady, linear trend for increasing number of herbicide area-treatments (simple linear regression $P$ value $< 0.001$, Fig. 1) over the last 25 years was observed for all crops except soybean. The linear trend was not statistically significant for soybean $(P = 0.271)$; it was instead characterized by a sustained decrease in the number of herbicide area-treatments between 1994 and 2005, followed by a marked increase between 2005 and 2015. Of the five crops characterized by a linear trend, herbicide use increased faster in rice (slope = 0.07), spring wheat (slope = 0.09) and winter wheat (slope = 0.06) compared with crops where glyphosate-resistant cultivars are widely planted (maize and cotton, slope = 0.05).

**Herbicide toxicity.** Chronic and acute toxicity values ranged widely between herbicides (Fig. 2). Chronic NOEL values ranged from 0.03 to 20,000 $mg\,kg^{-1}\,d^{-1}$, and acute $LD_{50}$ values ranged from 112 to 9,000 $mg\,kg^{-1}$. For acute toxicity, many herbicide active ingredients had $LD_{50}$ values listed as $> 5,000\,mg\,kg^{-1}$. EPA registration requirements state that pesticides with acute oral $LD_{50}$ values greater than 5,000 $mg\,kg^{-1}$ are considered Category IV, which is the least toxic category (40 CFR Ch I 156.62). Therefore toxicity tests to identify $LD_{50}$ values greater than this upper limit are unwarranted from the registrants' perspective. This censoring of acute toxicity values may result in slight bias in the acute toxicity data, since 5,000 $mg\,kg^{-1}$ was used as a conservative estimate for any herbicide where the $LD_{50}$ was listed as $> 5,000\,mg\,kg^{-1}$.

Pesticide toxicity is often discussed in a very general sense (e.g., using 'more toxic herbicides'), but there is not necessarily a strong relationship between acute and chronic toxicity, and therefore, the distinction between these two measures of toxicity is important. For the 118 active ingredients in these data, the correlation between chronic and acute toxicity values was not statistically significant (Pearson correlation $r = 0.096$, $P = 0.31$).

A hazard quotient approach[15] was used to evaluate the relative toxicity of herbicides being used in each crop over time. While the term 'hazard quotient' may not be familiar to many scientists, this approach has been used regularly in the literature, though not always identified by that name, to compare the relative toxicity of accumulated pesticides, herbicides and other toxins[16–18]. In the hazard quotient approach, the toxicity of a pesticide represents the hazard, and the amount of pesticide applied represents an estimate of exposure, so that the resulting hazard quotient provides an estimate of risk. The hazard quotient has a direct interpretation as the number of $LD_{50}$ or NOEL values applied per

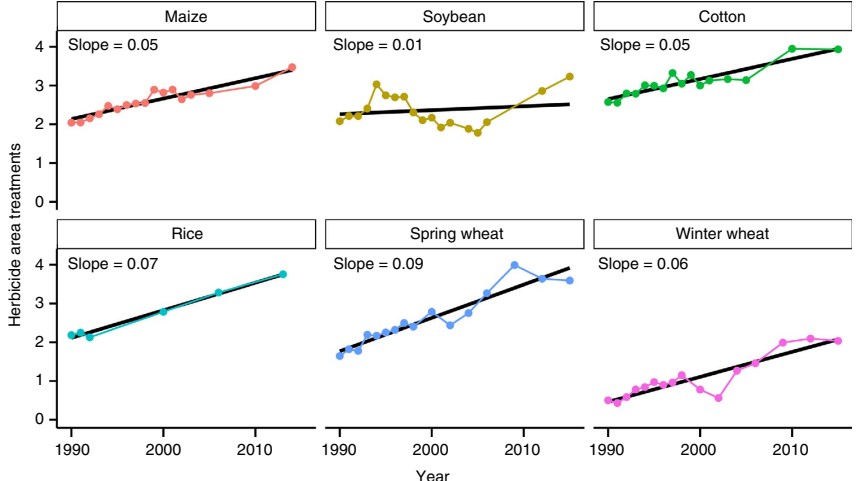

**Figure 1 | Herbicide area-treatments for six crops in the United States 1990 to 2015.** Area-treatments are an estimate of the number of herbicide treatments applied to each field. Linear regression $P$ value for soybean = 0.271; all other crops $P < 0.001$.

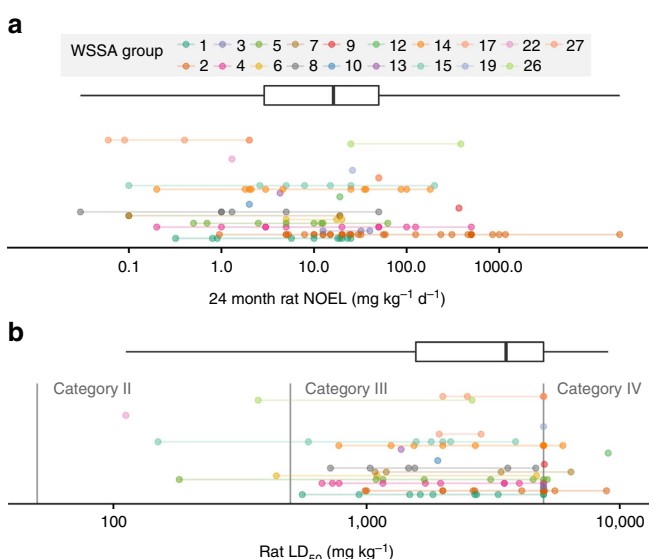

**Figure 2 | Toxicity of herbicides used between 1990 and 2015 in the United States.** Chronic toxicity (**a**) represented by 24-month oral rat no observable effect level (NOEL); acute toxicity (**b**) represented by oral rat LD$_{50}$. Toxicity is presented on a log$_{10}$ scale. For boxplots, bold vertical line represents median toxicity value, box encloses the inter-quartile range, and lines extend from minimum to maximum values. Each point represents a unique herbicide active ingredient, colour coded by WSSA site of action group number. Acute toxicity categories are those defined by US EPA.

hectare. High hazard quotient values indicate a relatively more toxic combination of herbicides.

**Maize toxicity trends.** The chronic hazard quotient has increased 7% in maize, from 1.57 million in 1990 to 1.68 million in 2014, though it has trended downward slightly in recent years (Fig. 3). Throughout the 1990's, atrazine was responsible for a large majority of the chronic hazard quotient in maize (Supplementary Fig. 1). In 2014, just two herbicides (atrazine and mesotrione) were responsible for 88% of the chronic hazard quotient. Acute herbicide toxicity has decreased 88% in maize, from an acute hazard quotient of 7016 in 1990 to 819 in 2014 (Fig. 4). Much of the reduction in acute toxicity was due to phasing out of alachlor

and cyanazine from the maize market. In 1990, alachlor and cyanazine accounted for 85% of the total acute hazard quotient (Supplementary Fig. 1).

**Soybean toxicity trends.** Chronic and acute herbicide toxicity in soybean has decreased 78% and 68%, respectively, between 1990 and 2015 (Figs 3 and 4). Most of the reduction in the chronic hazard quotient has been due to reduction in linuron use, while most of the acute hazard quotient reduction was due to reduction in alachlor use (Supplementary Fig. 2). In 1990, linuron was responsible for 80% of the chronic hazard quotient, and alachlor was responsible for 79% of the acute hazard quotient in soybean. In 2015, paraquat was responsible for 25% of the acute hazard quotient in soybean. In 2005, which was the peak of glyphosate dominance in the soybean market in this USDA-NASS data set, glyphosate represented 76% of all area-treatments, but was responsible for 10 and 75% of chronic and acute toxicity, respectively.

**Cotton toxicity trends.** The chronic hazard quotient for cotton increased between 1990 and 2015 (Fig. 3), although the increase has been driven almost completely by a single herbicide. In 2015, diuron was responsible for 89% of the chronic hazard quotient in cotton (Supplementary Fig. 3). The acute hazard quotient has decreased 65% from a peak of 1654 in 1994 to a low of 583 in 2003. After 2004, acute toxicity increased to 934 by 2015, but that was still substantially lower than any acute hazard quotient value observed before 2001. Much of the reduction in cotton acute hazard quotient was due to phasing out of the herbicide cyanazine, which made up 60% to 70% of the acute hazard quotient between 1990 and 1998. Similar to soybean, the peak of glyphosate dominance in the cotton market occurred in 2005 in this USDA-NASS data set, when glyphosate represented 54% of all area-treatments. Even with this high reliance on glyphosate for weed control, this herbicide was responsible for only 0.2% of the chronic hazard quotient. Glyphosate's contribution to the acute hazard quotient was similar to its contribution to total area-treatments, at 52% of acute toxicity in 2005.

**Rice toxicity trends.** Herbicide use in rice was only surveyed six times over the last 25 years, but since the surveys were conducted near the beginning and end of the period they still provide valuable information on the trend in herbicide use. In 1990, the chronic hazard quotient for rice (50.8 million) was far greater

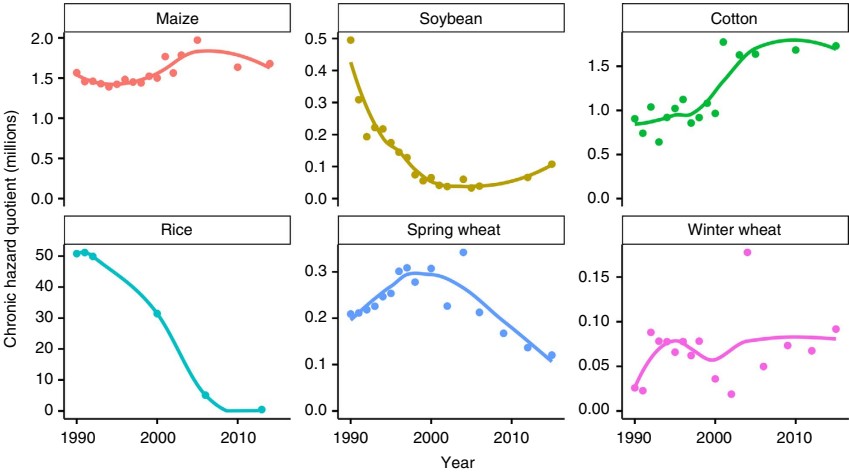

**Figure 3 | Herbicide chronic hazard quotients for six crops 1990 to 2015.** Hazard quotient is the number of chronic rat (oral, 24 month) NOEL values applied per hectare. Mann-Kendall tests for monotonic trend: maize (tau = 0.50, P = 0.006); soybean (tau = −0.69, P < 0.001); cotton (tau = 0.60, P = 0.001); rice (tau = −0.87, P = 0.024); spring wheat (tau = 0.02, P = 0.964); winter wheat (tau = 0.10, P = 0.620).

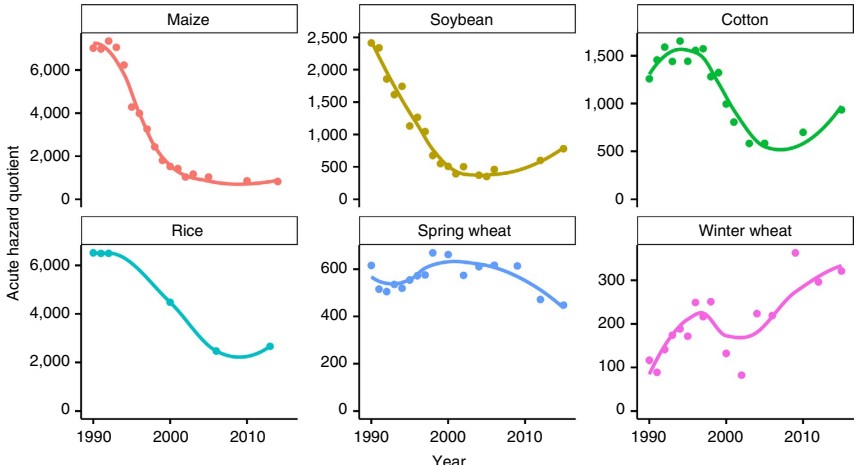

**Figure 4 | Herbicide acute hazard quotients for six crops 1990 to 2015.** Hazard quotient is the number of acute rat (oral) $LD_{50}$ values applied per hectare. Mann-Kendall tests for monotonic trend: maize (tau = −0.93, P < 0.001); soybean (tau = −0.71, P < 0.001); cotton (tau = −0.52, P = 0.006); rice (tau = −0.73, P = 0.060); spring wheat (tau = 0.13, P = 0.500); winter wheat (tau = 0.53, P = 0.005).

than any other crop in this analysis (Fig. 3), but was almost completely driven by a single herbicide. Molinate made up 99% of the chronic hazard quotient in 1990 (Supplementary Fig. 4). Although molinate use (and the associated chronic hazard quotient) declined dramatically between 1990 and 2006, molinate still accounted for 93% of the chronic hazard quotient in 2006. US registration of molinate herbicide was cancelled in 2008, and further use of molinate was prohibited after 2009 (ref. 19). In 2013, after molinate use was discontinued, thiobencarb and propanil made up 56 and 25% of the chronic hazard quotient in rice, respectively.

Molinate was also a substantial contributor to the acute hazard quotient, accounting for 32% of herbicide acute toxicity in 1990 (Supplementary Fig. 4). Discontinuation of molinate, therefore, also had a beneficial impact on acute toxicity of rice herbicides. Propanil has been the largest contributor to the acute hazard quotient over time in rice, accounting for 58% and 75% of the acute hazard quotient in 1990 and 2013, respectively.

**Spring wheat toxicity trends**. The chronic hazard quotient in spring wheat decreased 42% from 210,000 in 1990 to 120,000 in

2015 (Fig. 3). MCPA and 2,4-D accounted for 56% and 20% of the chronic hazard quotient in 1990, respectively, compared with 60% and 11% in 2015 (Supplementary Fig. 5). The acute hazard quotient has remained relatively steady in spring wheat, although a decreasing trend is apparent since 1998 (Fig. 4). Bromoxynil accounted for the greatest proportion of the acute hazard quotient in 2015 at 46% of the total.

**Winter wheat toxicity trends**. The chronic hazard quotient for winter wheat has remained relatively flat (Fig. 3), with the exception of peaks resulting from diuron (in 1992 followed by a decline through 2004) and dichlorprop which was present only in a single year in 2004 (Supplementary Fig. 6). Dichlorprop was only observed in the NASS data set once (in 2004 in both spring and winter wheat). The acute hazard quotient has increased rather steadily from 116 to 321 in 2015 (Fig. 4). 2,4-D has been the most consistent contributor to acute hazard quotient throughout the last 25 years, though both bromoxynil and glyphosate have increased in recent years (Supplementary Fig. 6). Although acute toxicity increased in winter wheat, both chronic and acute hazard quotients were generally lower for winter wheat

than all other crops in this analysis, primarily because winter wheat also had the fewest area-treatments applied (Fig. 1).

## Discussion

Weeds are a fact of life for farmers around the world, and weeds influence many farming decisions either directly or indirectly. If left uncontrolled, weeds could reduce world food production by as much as 20–40% (ref. 20). To control the weeds and increase their marketable crop, farmers around the world have increasingly turned to herbicides. When viewed in isolation, the increase in herbicide reliance is troubling. Use of glyphosate herbicide in particular has received increased scrutiny due to its association with the most dominant GE crop trait. A dramatic increase in glyphosate use[21] has justifiably generated concern among scientists, policy-makers and the general public. As this analysis shows, however, the increased use of herbicides may not be inherently bad, as sometimes these changes corresponded with lower toxicity. This analysis provides only a small component of the potential impacts related to herbicide use, and does not account for risks to the environment (or any potential benefits).

A variety of risk assessment methods can be used to compare herbicides, including the 'risk cup' method used by the US Environmental Protection Agency and other regulatory bodies. Regulatory agencies typically consider a wide variety of environmental and human health endpoints in their risk analysis processes. Risk analysis is complex even when considering only a single active ingredient, since multiple endpoints must be considered (applicator health, aquatic organisms, birds, insects, etc.). Risk analysis becomes far more complex when looking at multiple herbicides used across multiple crops. The results of a full environmental analysis are likely to be similarly mixed since soil persistence, leaching potential, and wildlife toxicity of these 118 herbicides differ at least as much as mammalian toxicity.

To fully understand the impacts of herbicide use changes, meaningful metrics that represent actual risk must be used. Previous analyses have attempted to quantify the environmental and health impacts of herbicide use over time, especially as it relates to adoption of genetically-engineered (GE) herbicide-resistant crops. Unfortunately, many of those efforts relied on fundamentally flawed metrics. In particular, the summed weight of herbicides applied with no regard for their relative toxicity is uninformative at best and misleading at worst[7]. Simply counting the kg applied is insufficient. Herbicide use rates range from grams to kilograms per hectare and depend on many factors, including the effectiveness of the active ingredient and the environment where it is applied. A large increase in the weight of herbicide applied could simply be due to a switch from a herbicide which is active at low doses to a less bioactive herbicide. Likewise, a reduction in the total weight of herbicide applied may not actually be indicative of reduced herbicide use, as a single herbicide may be replaced by many different herbicides with lower use rates, and could actually pose substantially greater risk to applicators and the environment.

This analysis corrects this deficiency of previous works, by using area-treatments as a more informative indicator of herbicide intensity. An upward trend in herbicide area-treatments was observed in all six crops that were analyzed, although the upward trend was preceded by a downward trend in soybean. This result is consistent with the 'herbicide treadmill' criticism suggesting that US crop production has become increasingly dependent on herbicides for weed control. No causal relationships can be determined from these data, however, and there are many factors that may have driven increased herbicide use over time. Use of tillage in the US has steadily decreased in most crops since 1996, though the rate of tillage reduction depends on the crop and

growing region[22]. Whether or not tillage is used explicitly for weed control, most tillage operations will provide weed control benefits like killing emerged seedlings and burying weed seed. When tillage is reduced, farmers become more reliant on other weed control practices, including herbicides. At least some of the widespread increase in herbicide use is certainly attributable to adoption of conservation tillage practices. It is important, then, to weigh the concern of increased herbicide use with the benefits that may have also accrued.

Although no new major herbicide sites of action have been discovered in the last 25 years[23], many new herbicide products have entered the market. Many of these new products contain multiple active ingredients. Increased marketing and use of these multi-ingredient products may have contributed to increased herbicide area-treatments, though this data set did not provide commercial formulation information so it is unclear whether this was the case.

Some researchers have blamed glyphosate-resistant crops and the resulting evolution of glyphosate-resistant weeds for increasing herbicide use in maize, soybean, and cotton[2,6]. While this explanation is plausible for these three glyphosate-resistant crops, it cannot explain the similar trends for increasing herbicide intensity in rice and wheat, since no glyphosate-resistant cultivars are commercially available for those crops. In fact, herbicide area-treatments increased at a faster rate in rice and wheat compared with the glyphosate-resistant crops, so the claim that glyphosate-resistant crops are the primary driver of increasing herbicide use is at odds with the empirical data. The broader problem of herbicide-resistant weeds (rather than the artificially narrow focus on glyphosate) may certainly have played a role in increasing herbicide use for all of the crops in this analysis. The most likely explanation, though, is probably a combination of inter-related factors and is far more complex than any single driver.

The EIQ commonly used in previous analyses of herbicide use over time suffers from severe methodological flaws[12,24] that are even more pronounced when comparing herbicides[13]. The hazard quotient approach used here, while certainly not perfect, is a far more defensible metric with which to compare herbicide toxicity and relative impacts of herbicide use changes, albeit for a small subset of potential toxicity endpoints. The hazard quotient, as applied here, does not take into account potential interactions between multiple herbicides. As an increasing number of herbicides are applied per hectare, the risk of negative interactions necessarily increases, although there is little evidence to date that negative interaction effects are of major concern to applicators or to the environment.

This analysis was limited to mammalian toxicity, and therefore is most relevant to chronic and acute risks faced by pesticide applicators, and to a much lesser extent, consumers. This analysis should not be extrapolated to draw conclusions about non-mammalian systems, and should be interpreted with caution even for human health risks. However, Peterson[25] demonstrated that lower tier risk assessment approaches such as the hazard quotient used here are indeed correlated to more in-depth risk analyses, and therefore the hazard quotient results are likely representative of actual risk.

Acute herbicide toxicity is relatively simple to quantify and interpret, since the endpoint of interest in acute toxicity testing is mortality. To put it bluntly, it is simple to determine whether a rat is dead or alive. The herbicide dose resulting in death of 50% of test animals ($LD_{50}$) is a standard measure of acute toxicity, and is required as part of a standard set of pesticide safety studies to obtain regulatory approval. Chronic toxicity is more difficult to quantify and standardize, since the endpoint of interest can vary widely; liver deformations, cancers, reduced body weight, or any other departure from a healthy test population can indicate

chronic toxicity issues. Chronic studies also have greater variation in study design, test species, duration, and endpoints measured, adding to the complexity. When making pesticide registration decisions, a variety of chronic studies conducted on a variety of test organisms are evaluated in an attempt to determine the most relevant endpoints and to set residue tolerances, acceptable use rates, and acceptable daily intakes. This makes it somewhat difficult to make comparisons between herbicides with respect to chronic toxicity comparisons.

Of the chronic toxicity data that are readily available for herbicides, the no observable effect level (NOEL) from 24-month chronic rat studies is the most consistent, and was therefore chosen to compare the chronic toxicity of herbicides in this analysis. This choice has the benefit of allowing an 'apples to apples' comparison of various herbicide active ingredients, since the chronic studies were conducted on the same test species for the same amount of time. However, rat NOEL values do not necessarily relate directly to human health risk. For some chronic effects, the rat is not an ideal test model for humans, and rabbit or dog studies may provide results more relevant to applicator health risks. Selecting different test species for different herbicides would be a potential source of bias in this analysis, so the same test organism (rat) was used for all active ingredients.

Of particular note in these results is that acute toxicity hazard (which is commonly cited by proponents of GE technology) was not always similar to chronic toxicity results. For example, the acute toxicity hazard decreased in cotton while the chronic toxicity hazard increased. Overall, acute mammalian toxicity of herbicides used in the US has decreased over the last 20 to 25 years for four out of six crops, while chronic toxicity has decreased for two of the six crops. It is important to note that the Mann-Kendall statistical test in Figs 3 and 4 only evaluates monotonic trends over the entire 25 year period. In some cases, more recent trends may be important even where the overall trend is non-significant (e.g. chronic toxicity in spring wheat, Fig. 3), or may even be reversed compared with long-term trends (e.g. acute toxicity in cotton, Fig. 4). The largest decreases in both hazard quotients were a result of discontinuation of several products with relatively high toxicity including alachlor, cyanazine, and molinate. In this regard, the EPA's decisions to discontinue these products appear to have had a beneficial effect on applicator health risks.

Because adoption of genetically engineered (GE) herbicide-resistant crops was so rapid and so widespread, the temporal component confounds the ability to define causal relationships between adoption of GE crops and herbicide use trends described here. Brookes and Barfoot[10] convincingly explain that extrapolating recent non-GE herbicide usage to represent what all non-GE crop growers would be doing in the absence of GE technology is problematic for several reasons. The minority of growers not using GE technology today are probably not representative of all growers, and therefore their pesticide use is almost certainly not an accurate way to compare overall pesticide use between GE and conventional crops. For example, farmers might not adopt glyphosate-resistant crops because weed densities on their farm are relatively low, or if the farmer is not managing herbicide-resistant weeds. Herbicide use is likely to be lower for these non-adopters regardless of which technology they use for weed control. Results of these comparisons would likely bias results toward higher herbicide use in GE crops.

Increased use of glyphosate was an obvious result of US farmers adopting glyphosate-resistant maize, soybean, and cotton. Increased glyphosate use has spurred debate about the safety of glyphosate, with the World Health Organization International Agency for Research on Cancer (IARC) declaring that glyphosate is 'probably carcinogenic to humans'[26] while the US Environmental Protection Agency (EPA) recently concluded that glyphosate is 'not likely to be carcinogenic to humans at doses relevant to human health risk assessment'[27]. Neither the IARC nor EPA analyses assess whether glyphosate use is better or worse than herbicides (or other weed control strategies) that would be used in its place.

Although USDA data do not allow direct comparison between herbicide use in glyphosate-resistant versus conventional varieties, some general conclusions can be drawn in this regard. Glyphosate has an approximate acute $LD_{50}$ of $5,037\,mg\,kg^{-1}$, with some variation depending on which salt is applied. This makes glyphosate less acutely toxic than 94% of the herbicides in this data set. Although glyphosate is considered a relatively safe herbicide with respect to acute toxicity, it is not an outlier in this regard. The median acute $LD_{50}$ for herbicides in this analysis was $3,556\,mg\,kg^{-1}$, and only five herbicides had acute $LD_{50}$ of less than $500\,mg\,kg^{-1}$, placing them in EPA's toxicity Category II (Fig. 2). Therefore, the contribution of glyphosate to acute toxicity was nearly the same as its contribution to herbicide use as measured by area-treatments; that is, if glyphosate made up 20% of area-treatments, it typically contributed to just under 20% of the acute hazard quotient.

Chronic toxicity was a different story, however. Glyphosate has a lower chronic toxicity than 90% of all herbicides in this analysis, but it falls much further from the median chronic toxicity value compared with acute toxicity (Fig. 2). In the last year of survey data for each crop, glyphosate made up 26% of maize, 43% of soybean, and 45% of cotton area-treatments, but only contributed 0.1%, 0.3% and 3.5% of the total chronic hazard quotients in those crops, respectively. So although the chronic hazard quotient increased in 2 of 3 glyphosate-resistant crops, if glyphosate were not used the chronic hazard quotient would almost certainly be even greater since other herbicides with greater chronic toxicity would have been used instead. Similarly, if glyphosate use were discontinued (as was recently proposed in the EU) the resulting displacement of glyphosate by other herbicides is likely to have a negative impact on chronic health risks faced by pesticide applicators[28].

## Methods

**Data sources.** Data for herbicide use and crop planted area were downloaded from USDA-NASS (quickstats.nass.usda.gov) for all available years between 1990 and 2015 (provided as Supplementary Data Sets 1 through 7). For each herbicide active ingredient included in the NASS data, the herbicide site of action (by WSSA code), the acute rat $LD_{50}$, and the chronic 24 month rat NOEL was recorded. Site of action and toxicity data were collected from the Herbicide Handbook[29] if available, otherwise US EPA registration documents were searched to find the information.

**Area-treatments.** Heeding the recommendation from the recent National Academies report[7], total herbicide applied in kg of active ingredient per hectare is not presented or discussed in this report. Instead, area-treatments were calculated. The total amount of each herbicide active ingredient applied per crop per year was divided by the average application rate (Rate) within each crop for each year, then further divided by the number of planted acres (acres) of that crop in that year to obtain area-treatments (AT).

$$AT = \frac{Amount/Rate}{acres} \qquad (1)$$

All area-treatments from Equation (1) were then summed for each herbicide ai to determine the total number of area-treatments applied in each year to each crop.

**Relative toxicity.** For this analysis, the hazard quotient (HQ) is defined as the sum of the amount of each herbicide applied per hectare divided by the toxicity of each herbicide (equation (2)):

$$HQ = \sum_{ai=1}^{N} \frac{Amount_{ai}}{Toxicity_{ai}} \qquad (2)$$

where $N$ is the total number of herbicide active ingredients applied to a crop in a year, Amount is the total weight of each active ingredient (ai) applied in $mg\,ha^{-1}$, and Toxicity is either the chronic or acute toxicity value for each ai. For the chronic

hazard quotient, Toxicity is the 24 month rat NOEL expressed in $mg\,kg^{-1}$ body weight $d^{-1}$. For the acute hazard quotient, Toxicity is the acute rat $LD_{50}$ expressed in $mg\,kg^{-1}$.

**Software.** Data analysis and figure construction were done using the *R* statistical language, and relied heavily on the *dplyr*[30], *tidyr*[31], *ggplot2* (ref. 32), and *cowplot*[33] packages.

**Data availability.** All data used in the analysis have been provided as Supplementary Information.

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

## Acknowledgements

Publication costs were provided by the Wyoming Agricultural Experiment Station through funding received by United States Department of Agriculture National Institute for Food and Agriculture, Hatch Project WYO-528-14.

## Author contributions

A.R.K. conceived, designed, and conducted the analysis, created the figures, and wrote the manuscript.

## Additional information

**Competing interests:** No specific funding was received related to this manuscript. Funding has been provided to the University of Wyoming from the following sponsors in support of Dr Kniss's research and education program, either through unrestricted gifts, research contracts, or grants: Arysta LifeScience, BASF, Bayer CropScience, Dow AgroSciences, DuPont, FMC, Hatch Act Funds–USDA, Loveland Industries, Monsanto, NovaSource, Repar Corporation, StateLine Bean Cooperative, Syngenta, USDA National Institute for Food and Agriculture, University of Wyoming Department of Plant Sciences, University of Wyoming School of Energy Resources, Valent, Western Sugar Cooperative, Winfield Solutions, Wyoming Agricultural Experiment Station, Wyoming Crop Improvement Association, Wyoming Department of Agriculture, and Wyoming Seed Certification.

