## [Peer Review File · Nature Communications]

Reviewers' comments:

Reviewer #1 (Remarks to the Author):

The manuscript adds a novel analysis and important information on trends in, and impact of, herbicide use. This analysis is needed because of the myriad ways these analyses can be conducted (and spun by those with vested interests). The manuscript should be of considerable interest to researchers and others in several disciplines. The manuscript is well written and there are only minor suggestions for improvement (see below).

The author should stress even more the disadvantages of non-risk assessment approaches (i.e., indexes and approaches that do not rely on estimating the joint probability of toxicity and exposure). Risk assessment is the ideal method for estimating the environmental effect of herbicide use, but the information and expertise required often prevents its use for questions that are the topic of the author's manuscript.

The author should also see and cite (if appropriate): Peterson, R.K.D. 2006. Comparing ecological risks of pesticides: the utility of a risk quotient ranking approach across refinements of exposure. *Pest Management Science* 62: 46-56. The paper supports the author's approach, in a sense, by showing that that numerical ranking of risk quotients for the purpose of comparing potential risks is a valid approach because the rankings are significantly correlated regardless of the degree of exposure refinement. This finding could support the author's approach of using herbicide use intensity as a surrogate for exposure because Peterson (2006) show that lower tiers of assessment are correlated to higher tiers. If the author's placements of hazard quotients reflect placements at higher tiers of exposure refinement, his conclusions could be more broadly applicable to risk.

L7. Delete the hyphen after "genetically".

L10 (and throughout). Replace "corn" with "maize".

L11. Replace "blamed" for another term that is less judgmental and more appropriate to a scientific article.

L17 (and throughout, where appropriate). Replace "it's" with "its".

L30. Need citations for the statement, "There is increasing public interest in how herbicide use has changed over time..."

L68. Replace "is" with "are".

L90. Replace "is" with "are".

L102. Delete "in order".

L176. Replace "it's" with "its".

L217. Replace "it's" with "its".

L240. Replace "this" with "these".

L270. Add the citation to the Dushoff et al citation: Peterson, R.K.D., and J.J. Schleier III. 2014. A probabilistic analysis reveals fundamental limitations with the environmental impact quotient and similar systems for rating pesticide risks. *PeerJ* 2:e364; DOI 10.7717/peerj.364.

L289. Delete the hyphen after "genetically".

L302. Replace "does" with "do".

L321-323. There are a few papers that support this statement. They should be cited here. Here are two, but there are others that are even more pertinent to the statement: Peterson, R.K.D. 2006. Comparing ecological risks of pesticides: the utility of a risk quotient ranking approach across refinements of exposure. *Pest Management Science* 62:46-56; Peterson, R.K.D., and A.G. Hulting. 2004. A comparative ecological risk assessment for herbicides used on spring wheat: the effect of glyphosate when used within a glyphosate-tolerant wheat system. *Weed Science* 52:834-844.

Reviewer #2 (Remarks to the Author):

General Comments

The author makes two major claims. First, in the USA from 1990 to 2015, herbicide use intensity in non-glyphosate-resistant wheat and rice has increased more rapidly than in glyphosate-resistant corn, soybean and cotton. Second, in the USA from 1990 to 2015, although herbicide use intensity has increased, relative toxicity hazards have decreased. Both claims are novel, convincing and are based on credible data sets available to the general public.

It is very important to present herbicide use data as an estimate of the number of field applications (as in the current article); this metric avoids confounding total active ingredient and total biological activity issues that plague many herbicide use articles. The finding that herbicide use intensity in non-glyphosate-resistant crops has increased more rapidly than in glyphosate-resistant crops is very important to publish and provides insight into herbicide use patterns in cropping systems that utilize different biotechnology levels in important world crops. Given widely disparate opinions regarding current herbicide use intensity in genetically modified versus non-genetically modified crops and their potential impact on public policy decisions, the herbicide use metric employed here is critical for new thinking on herbicide use intensity questions.

The finding that relative toxicity hazards have decreased despite increased herbicide usage will be of major interest in the field of weed science. It may not, however, change much thinking in that discipline; many already feel that glyphosate and some other relatively new herbicides are less toxic than those they have replaced. However, as a result of the article, many outside of weed science may change their thinking regarding the toxicity of major herbicides such as glyphosate. The limitation (as noted by the author) is that LD50s and NOELs from studies with rats are very narrow predictors of overall toxicity hazards to humans. This article will spark debate on whether or not similar predictors of human toxicity hazards will lead to similar conclusions. In the interest of making the article more palatable for broad readership, I think it is important to state in the discussion section that not all research supports glyphosate safety in humans. I suggest that the author cite at least one paper that raises concerns about glyphosate safety (e.g., Richard et al. 2005) and one that supports safety claims (e.g., Williams et al. 2000); there may be more appropriate papers .

Specific Comments

32, 64, 66-67, 106, 110-111, 191, 283 (Line numbers): Replace "herbicide programs" with "herbicides" throughout the article. "Herbicide programs" is a rather colloquial weed scientist jargon term that usually refers to a treatment regime that may include many herbicides or repeated applications of the same herbicide for an entire crop year. In most cases the article is about individual herbicides as opposed to entire herbicide regimes.

68: I suggest the sentence be revised as follows: "...scaled in EIQ calculations, it readily..."

71: The Kniss citation is "2016" in the References section.

99: I suggest the sentence be revised as follows: "...analysis so the same..."

101: I suggest the sentence be revised as follows: "...human health, weed resistance to herbicides, environmental..."

109: Replace "called" with "identified".

224: I suggest the sentence be revised as follows: "...mixed since soil..."

238: Insert "suggesting" after "criticism".

250: Insert "major" after "new" and provide the supporting reference for this statement (Duke 2012 Pest Manag Sci 68:505-512).

315: Insert "(Figure 2)" after the second "toxicity".

323: Insert "by" after "faced".

Figures S5-S10: In these Figure captions, the "A" appears to refer to the acute hazard quotient chart and the "B" appears to refer to the chronic hazard quotient chart. In the actual Figure the "A" chart is the chronic hazard quotient chart and the "B" chart is the acute hazard quotient chart.

K. Neil Harker

Reviewer #3 (Remarks to the Author):

This manuscript reports an analysis of trends in herbicide use 1990-present from USDA surveys.

A synthesis on this topic is certainly timely – for example the New York Times recently ran a front page article on GM crops in which the use of herbicides featured prominently (http://www.nytimes.com/2016/10/30/business/gmo-promise-falls-short.html?_r=0). Like the reviews of pesticide efficacy that seem to appear about once every ten years (Oerke, 2006. Crop Losses to Pests. *Journal of Agricultural Science*. 144: 31-43), a review/synthesis of this topic is likely to be very widely cited.

Overall, I think the data in Figs 1-4 are very important and they will be widely used. My suggestions for improvement focus on shortcomings in presentation and discussion, as follows.

1. The first thing I did was to draw the trends for acute and chronic hazard quotients (Figs 3 and 4) onto the same pair of axes for each crop. When this is done, we observe that the trend lines match qualitatively in soybean, rice, and wheats and are discordant in maize and cotton.

In the discussion of the present paper, the author has largely re-described the figures list-wise for each crop with notes on major contributors to the trends (e.g. effect of molinate use on HQ trend in rice, paraquat in maize, etc.). For me, the first issue is to decide what it means when acute and chronic HQs match or don't match. I suggest dealing with soy-rice-wheat (matches) and maize-cotton in two successive sections to produce a truly synthetic discussion.

2. The Introduction doesn't seem to set objectives for the paper – there's a focus on justifying the summative-HQ approach over others that continues into the first part of the Discussion (lines 212-235 are essentially obsolete and discussion proper begins on 236).

In actuality, the paper ends up, quite rightly, with a detailed discussion of the controversial glyphosate issue, but this needed to be set up as an objective early on with some context.

3. I was initially confused by the term 'area-treatments' – it turned out to be 'applications'. Even if area-treatment is maintained, it would be helpful to relate it to the usual 'application'. Perhaps there's a need to explain how the problematic 'tank mix' issue (farmers often apply multiple chemicals at once) might have affected (or not) this calculation.

4. I am not convinced by the curve fitting used in Figs 3 & 4. Essentially, the central question is 'increase-vs-decrease' and so fitting lines that wander like a moving average (e.g. acute HQ winter wheat) when the long term trend is flat 1995-2005 seems to blur rather than clarify the issue at hand

All changes that were made exactly as suggested are simply marked with 'Δ'. Explanations are provided where appropriate.

Reviewer #1 (Remarks to the Author):

The manuscript adds a novel analysis and important information on trends in, and impact of, herbicide use. This analysis is needed because of the myriad ways these analyses can be conducted (and spun by those with vested interests). The manuscript should be of considerable interest to researchers and others in several disciplines. The manuscript is well written and there are only minor suggestions for improvement (see below).

The author should stress even more the disadvantages of non-risk assessment approaches (i.e., indexes and approaches that do not rely on estimating the joint probability of toxicity and exposure). Risk assessment is the ideal method for estimating the environmental effect of herbicide use, but the information and expertise required often prevents its use for questions that are the topic of the author's manuscript. **Δ Added Something in the introduction, but I do feel this has been pretty well noted in a few places. In fact, Reviewer #3 felt this point may have actually been too prominent. So I have not made a major change in response to this suggestion.**

The author should also see and cite (if appropriate): Peterson, R.K.D. 2006. Comparing ecological risks of pesticides: the utility of a risk quotient ranking approach across refinements of exposure. *Pest Management Science* 62:46-56. The paper supports the author's approach, in a sense, by showing that that numerical ranking of risk quotients for the purpose of comparing potential risks is a valid approach because the rankings are significantly correlated regardless of the degree of exposure refinement. This finding could support the author's approach of using herbicide use intensity as a surrogate for exposure because Peterson (2006) show that lower tiers of assessment are correlated to higher tiers. If the author's placements of hazard quotients reflect placements at higher tiers of exposure refinement, his conclusions could be more broadly applicable to risk. **Δ Added information to the manuscript (near end of Introduction and middle of Discussion sections) in response to this comment.**

L7. Delete the hyphen after "genetically". Δ

L10 (and throughout). Replace "corn" with "maize". Δ

L11. Replace "blamed" for another term that is less judgmental and more appropriate to a scientific article. Δ

L17 (and throughout, where appropriate). Replace "it's" with "its". Δ

L30. Need citations for the statement, "There is increasing public interest in how herbicide use has changed over time..." **Δ sentence changed to remove "public interest" so that a citation is not necessary.**

L68. Replace "is" with "are". Δ

L90. Replace "is" with "are". Δ

L102. Delete "in order". Δ

L176. Replace "it's" with "its". Δ

L217. Replace "it's" with "its". Δ

L240. Replace "this" with "these". Δ

L270. Add the citation to the Dushoff et al citation: Peterson, R.K.D., and J.J. Schleier III.

2014. A probabilistic analysis reveals fundamental limitations with the environmental impact quotient and similar systems for rating pesticide risks. PeerJ 2:e364; DOI 10.7717/peerj.364. **▲**

L289. Delete the hyphen after “genetically”. **▲**

L302. Replace “does” with “do”. **▲**

L321-323. There are a few papers that support this statement. They should be cited here. Here are two, but there are others that are even more pertinent to the statement: Peterson, R.K.D. 2006. Comparing ecological risks of pesticides: the utility of a risk quotient ranking approach across refinements of exposure. Pest Management Science 62:46-56; Peterson, R.K.D., and A.G. Hulting. 2004. A comparative ecological risk assessment for herbicides used on spring wheat: the effect of glyphosate when used within a glyphosate-tolerant wheat system. Weed Science 52:834-844. **▲**

Reviewer #2 (Remarks to the Author):

General Comments

The author makes two major claims. First, in the USA from 1990 to 2015, herbicide use intensity in non-glyphosate-resistant wheat and rice has increased more rapidly than in glyphosate-resistant corn, soybean and cotton. Second, in the USA from 1990 to 2015, although herbicide use intensity has increased, relative toxicity hazards have decreased. Both claims are novel, convincing and are based on credible data sets available to the general public.

It is very important to present herbicide use data as an estimate of the number of field applications (as in the current article); this metric avoids confounding total active ingredient and total biological activity issues that plague many herbicide use articles. The finding that herbicide use intensity in non-glyphosate-resistant crops has increased more rapidly than in glyphosate-resistant crops is very important to publish and provides insight into herbicide use patterns in cropping systems that utilize different biotechnology levels in important world crops. Given widely disparate opinions regarding current herbicide use intensity in genetically modified versus non-genetically modified crops and their potential impact on public policy decisions, the herbicide use metric employed here is critical for new thinking on herbicide use intensity questions.

The finding that relative toxicity hazards have decreased despite increased herbicide usage will be of major interest in the field of weed science. It may not, however, change much thinking in that discipline; many already feel that glyphosate and some other relatively new herbicides are less toxic than those they have replaced. However, as a result of the article, many outside of weed science may change their thinking regarding the toxicity of major herbicides such as glyphosate. The limitation (as noted by the author) is that LD50s and NOELs from studies with rats are very narrow predictors of overall toxicity hazards to humans. This article will spark debate on whether or not similar predictors of human toxicity hazards will lead to similar conclusions. In the interest of making the article more palatable for broad readership, I think it is important to state in the discussion section that not all research supports glyphosate safety in humans. I suggest that the author cite at least one paper that raises concerns about glyphosate safety (e.g., Richard et al. 2005) and one that supports safety claims (e.g., Williams et al. 2000); there may be more appropriate papers. **▲ I have added a paragraph to the Discussion section comparing the IARC and EPA assessments of glyphosate, since these analyses are much more well-known and robust compared to**

any single study that could be cited (including the two suggested by the reviewer).

Specific Comments

32, 64, 66-67, 106, 110-111, 191, 283 (Line numbers): Replace “herbicide programs” with “herbicides” throughout the article. “Herbicide programs” is a rather colloquial weed scientist jargon term that usually refers to a treatment regime that may include many herbicides or repeated applications of the same herbicide for an entire crop year. In most cases the article is about individual herbicides as opposed to entire herbicide regimes. ▲

68: I suggest the sentence be revised as follows: “...scaled in EIQ calculations, it readily...” ▲

71: The Kniss citation is “2016” in the References section. ▲

99: I suggest the sentence be revised as follows: “...analysis so the same...” ▲

101: I suggest the sentence be revised as follows: “...human health, weed resistance to herbicides, environmental...” ▲

109: Replace “called” with “identified”. ▲

224: I suggest the sentence be revised as follows: “...mixed since soil...” ▲

238: Insert “suggesting” after “criticism”. ▲

250: Insert “major” after “new” and provide the supporting reference for this statement (Duke 2012 Pest Manag Sci 68:505-512). ▲

315: Insert “(Figure 2)” after the second “toxicity”. ▲

323: Insert “by” after “faced”. ▲

Figures S5-S10: In these Figure captions, the “A” appears to refer to the acute hazard quotient chart and the “B” appears to refer to the chronic hazard quotient chart. In the actual Figure the “A” chart is the chronic hazard quotient chart and the “B” chart is the acute hazard quotient chart. ▲

K. Neil Harker

Reviewer #3 (Remarks to the Author):

This manuscript reports an analysis of trends in herbicide use 1990-present from USDA surveys.

A synthesis on this topic is certainly timely – for example the New York Times recently ran a front page article on GM crops in which the use of herbicides featured prominently

(http://www.nytimes.com/2016/10/30/business/gmo-promise-falls-short.html?_r=0). Like the reviews of pesticide efficacy that seem to appear about once every ten years (Oerke, 2006. Crop Losses to Pests. Journal of Agricultural Science. 144:31-43), a review/synthesis of this topic is likely to be very widely cited.

Overall, I think the data in Figs 1-4 are very important and they will be widely used. My suggestions for improvement focus on shortcomings in presentation and discussion, as follows.

1. The first thing I did was to draw the trends for acute and chronic hazard quotients (Figs 3 and 4) onto the same pair of axes for each crop. When this is done, we observe that the trend lines match qualitatively in soybean, rice, and wheats and are discordant in maize and cotton.

In the discussion of the present paper, the author has largely re-described the figures list-wise for each crop with notes on major contributors to the trends (e.g. effect of molinate use on HQ trend in rice, paraquat in maize, etc.). For me, the first issue is to decide what it means when acute and chronic HQs match or don't match. I suggest dealing with soy-rice-wheat (matches) and maize-cotton in two successive sections to produce a truly synthetic discussion. **Δ The lack of strong relationship between chronic and acute toxicity is already discussed in the manuscript (3rd & 4th paragraph of the Results section), and Figure 2 illustrates that there is no strong relationship between the acute and chronic toxicity values. While I can certainly see the benefit of re-organizing the manuscript in this way, I think it draws attention away from the glyphosate-resistant vs conventional distinction (the current order of presentation). Therefore, I respectfully disagree that this suggestion would be helpful and have left the organization in the order originally presented.**

2. The Introduction doesn't seem to set objectives for the paper – there's a focus on justifying the summative-HQ approach over others that continues into the first part of the Discussion (lines 212-235 are essentially obsolete and discussion proper begins on 236).

In actuality, the paper ends up, quite rightly, with a detailed discussion of the controversial glyphosate issue, but this needed to be set up as an objective early on with some context. **Δ The final paragraph of the introduction (third sentence) explicitly stated the objectives of the paper. Though I agree with the reviewer that it was somewhat buried within the text. I have made this objective statement the final sentence of that paragraph, and moved the hazard quotient description to the Results section.**

3. I was initially confused by the term 'area-treatments' – it turned out to be 'applications'. Even if area-treatment is maintained, it would be helpful to relate it to the usual 'application'. Perhaps there's a need to explain how the problematic 'tank mix' issue (farmers often apply multiple chemicals at once) might have affected (or not) this calculation. **Δ I agree with the reviewer that the concept of area-treatments is defined too late in the manuscript; I have moved several sentences from the Methods section at the beginning of the area-treatment section of the Results so that this concept (and how it is derived and interpreted) is presented more clearly and readers don't have to flip back to the Methods to figure it out.**

4. I am not convinced by the curve fitting used in Figs 3 & 4. Essentially, the central question is 'increase-vs-decrease' and so fitting lines that wander like a moving average (e.g. acute HQ winter wheat) when the long term trend is flat 1995-2005 seems to blur rather than clarify the issue at hand. **Δ The curve fitting is a simple LOESS regression. The trends are not linear, so linear regression would be inappropriate. Also, there is enough 'float' from year to year that simply connecting the dots would be potentially misleading. I feel that the LOESS fit is the most appropriate and clearest way to show the 'trend' in the data rather than focusing on year to year variability.**

REVIEWERS' COMMENTS:

Reviewer #1 (Remarks to the Author):

The author has made revisions to my satisfaction.

Reviewer #2 (Remarks to the Author):

The author has appropriately addressed all of my concerns and suggested revisions.

K. Neil Harker

Reviewer #3 (Remarks to the Author):

Technically, there's nothing critical to remedy, but I'll put my three points again with as view to perhaps eliciting some minor adjustments.

1. Since the review of glyphosate and the comparison of GE/conventional agriculture are a major components of the paper, it would help publicise the broader relevance of the paper early on if they were integrated into the paper's objectives.
2. I agree with the author that the lack of correlation between acute/chronic toxicities accommodates the discordance between these two indices of chronological changes in maize/cotton. But this point is tacit in the paper; best to deal with it explicitly in the discussion.
3. While the LOESS approach is perfectly defensible for curve-fitting, how about adding a formal hypothesis test of overall trends with rank correlation? This allows one to make a conclusion about an overall trend for each graph across the time period without regard to its precise shape. Could be added parenthetically in, for example, the figure legends.

REVIEWERS' COMMENTS:

Reviewer #1 (Remarks to the Author):

The author has made revisions to my satisfaction.

Thank you.

Reviewer #2 (Remarks to the Author):

The author has appropriately addressed all of my concerns and suggested revisions.

K. Neil Harker

Thank you.

Reviewer #3 (Remarks to the Author):

Technically, there's nothing critical to remedy, but I'll put my three points again with as view to perhaps eliciting some minor adjustments.

1. Since the review of glyphosate and the comparison of GE/conventional agriculture are a major components of the paper, it would help publicise the broader relevance of the paper early on if they were integrated into the paper's objectives.

I have added GE vs non-GE crops to the objectives statement at the end of the introduction.

2. I agree with the author that the lack of correlation between acute/chronic toxicities accommodates the discordance between these two indices of chronological changes in maize/cotton. But this point is tacit in the paper; best to deal with it explicitly in the discussion.

I have added 2 sentences in the Discussion section to explicitly draw attention to this point.

3. While the LOESS approach is perfectly defensible for curve-fitting, how about adding a formal hypothesis test of overall trends with rank correlation? This allows one to make a conclusion about an overall trend for each graph across the time period without regard to its precise shape. Could be added parenthetically in, for example, the figure legends.

I have looked into some non-parametric tests for trend that may be appropriate here (including rank correlation). The most relevant (and commonly used) appears to be the Mann-Kendall test, which is often used to test for trends in environmental data sets. However, the Mann-Kendall test is only appropriate for monotonic trends (that is, continually increasing or continually decreasing) in time-series. Therefore, it is likely to be misinterpreted or over-interpreted when applied to the current data set. The trends observed are *not* all monotonic. For example, soybean area-treatments increase, then decrease, then increase again, and a test for monotonic trend would therefore have the same problems as a linear trend analysis: it would suggest that an overall trend is "not significant" (Mann-Kendall P-value = 0.40). But this conclusion would actually be inconsistent with the actual data and its interpretation because

the trend is not monotonic. The decrease and subsequent increase (described well by the loess regression) in herbicide use have plausible causes in the real-world (which are discussed in the manuscript). To say that these changes are statistically “not significant” because they are not monotonic would almost certainly be misinterpreted to mean the changes aren’t meaningful.

Additionally, a *significant* trend could also be misleading. For example, the Mann-Kendall P-value for acute cotton toxicity hazard is significant ($P=0.006$), indicating that acute toxicity hazard has decreased. However, the more recent loess trend shows that acute toxicity in cotton has increased since 2005, though with fewer data points in support due to USDA budget cuts. It wouldn’t be appropriate, in my opinion, to suggest that this recent increase in cotton acute toxicity hazard isn’t “significant” because a monotonic test shows the opposite result.

Therefore, I have included trend test statistics and p-values as requested by Reviewer 3 in Figures 3 & 4, but I have placed minimal focus on them in the text, and also added a cautionary note summarizing this response in the discussion section of the manuscript.